

# Hydrogels improved parsley (*Petroselinium crispum*(Mill.) Nyman) growth and development under water deficit stress

M. Zeki Karipçin

Faculty of Agriculture, Horticulture Department, Siirt University, Siirt, Turkey

## ABSTRACT

Water scarcity is one of the most pressing problems facing countries in the semi-arid and arid regions of the world. Data predicts that by 2030, global water consumption will increase by 50%, leading to severe water shortages. Today, agricultural production consumes more than 70% of fresh water in many parts of the world, increasing the pressure on water scarcity. For these reasons, agricultural production models and approaches should be developed to reduce water consumption. One developed approach is the use of hydrogel to reduce water consumption and have a positive effect on plant growth. This study investigated the use of hydrogels as chemical components that can be used in water shortage conditions and against the expected water scarcity. Parsley was used as the model organism. The method used was as follows: two different water treatments (50% and 100%) and four different hydrogel concentrations (0%, 50%, 75%, and 100%) were applied, and root width and length, leaf width and length, main stem length, and the number of tillers were measured. According to the results, while no improvement was observed in the plants with 100% hydrogel concentration, the best results were obtained from 50% hydrogel application. The results obtained from 75% hydrogel application were found to be higher than those of 100% hydrogel but lower than 0% hydrogel application. With 50% hydrogel (water-restricted), all plant growth parameters were higher compared to the plants with 100% (full irrigation) water application. It was determined that the average value of the I1 (50%) irrigation was the highest (3.6), and the average value of the I2 (100%) irrigation (2.4) was the lowest. It was determined that the highest average value (6.2) in all measured traits was the average value of the H1 (50%) application, and the lowest average value (0.0) was in the H3 hydrogel applications (100%). In conclusion, this study suggested that hydrogel application is beneficial on a large scale, can optimize water resource management for higher yields in agriculture, and has a positive effect on agricultural yield under water deficit stress.

Corresponding author
M. Zeki Karipçin,
zkaripcin@siirt.edu.tr

## INTRODUCTION

*Petroselinum crispum* is cultivated in temperate and subtropical climates worldwide and is often used as an aromatic herb in cooking, in traditional medicine for its active ingredient, and for landscaping. The parsley vegetable is cultivated for both its leaves (either flat or broad) and tuber (root). The main varieties of the parsley vegetable are Italian (or flat-leaf),

curly-leaf, and root parsley (*Miller et al., 2020*). *P. crispum* has green leaves that have a mild, pleasant flavor, and contain antioxidants, the flavonoid luteolin, and vitamins C and A. This vegetable is a source of iodine and iron and is rich in folic acid (*Stephens, 1994*; *Ajmera, Kalani & Sharma, 2019*). *P. crispum* has antioxidant properties, protects against DNA damage, and inhibits the proliferation and migration of cancer cells (*Tang et al., 2015*). However, the possible curative effects of hydrogels on parsley development have not been studied before.

Hydrogels are a type of gel obtained from the chemical synthesis of hydrophilic polymers (*Kabir, Ahmed & Furukawa, 2017*; *Qasim et al., 2018*). Hydrogels retain their structural integrity and absorb water from the environment (*Montesano et al., 2015*; *Domalik-Pyzik, Chłopek & Pielichowska, 2019*; *Qu & Luo, 2020*). Water hydrogels can be classified into three groups: (1) synthetic hydrogels that contain polyacrylic, polyacrylamide, and polyacrylonitrile; (2) semi-synthetic hydrogels that contain starch-polyacrylonitrile, starch-polyacrylamide, and starch polyacrylic acid; and (3) hydrogels such as cellulose and guar gum that originate from natural raw materials. The biggest limitations of the use of synthetic and semi-synthetic hydrogels in agriculture are their cost, environmental effects, and most importantly, non-biodegradability. Hydrogels obtained from natural substances have a short life span in soil, but synthetic or semi-synthetic hydrogels can work in soil for one or more generations of plants in a stable state.

More than 70% of the water used in agricultural irrigation is usable water. Some of the greatest advantages of hydrogels are that they can be obtained from natural wastes and their potential use in soil water conservation (*Sharma, Bahuguna & Dadrwal, 2021*; *Milijković, Gajić & Nikolić, 2021*). There have been only a few studies reporting the use of hydrogels in agriculture. *Wilske et al. (2014)* reported that the dissolution rate of polyacrylate hydrogel, a synthetic hydrogel, was just 0.45% over 24 weeks. The sodium alginate/polyacrylamide hydrogel developed by *Elbarbary, El-Rehim & El-Sawy (2017)* stimulated corn growth and sustained nutrient release into the soil.

A hydrogel developed from gum arabic by *Hasija, Sharma & Kumar (2018)* significantly increased soil's water holding capacity. It was reported that amino ethyl chitosan and acrylic acid obtained by free radical polymerization was able to hold water and release stored water into the soil, and performed excellently under salt and drought stress conditions (*Fang et al., 2018*). However, the high degradability of biogel is undesirable in soil applications (*Song et al., 2020*; *Dhanapal et al., 2021*). Hydrogels that are degradable in soil but are more stable and can be used in at least one or more plant growth cycles are therefore preferable in agricultural areas (*Kai et al., 2016*; *Hasija, Sharma & Kumar, 2018*). However, according to *Nyyssölä & Ahlgren (2019)*, diverse aerobic bacteria extracted from environments containing polyacrylamide and polyacrylate or from soil samples have been used to catabolize these polymers. Polyacrylates have been shown to be effective when exposed to controlled-release fertilizer (CRF), also known as coated fertilizer. Fourier transform infrared photoacoustic spectroscopy (FTIR-PAS) and laser-induced breakdown spectroscopy (LIBS) have been used specifically in the analysis of the *in-situ* surface in the degradation steps of the polymer (*Liang et al., 2018*).

Because food supply and demand issues, producers use more synthetic fertilizers (especially N, P, and K) per unit area, which has shaken the foundations of both sustainable and traditional agriculture. According to *Prakash, Kavitha & Maheshwari (2021)* and *Sabyasachi & Prakash (2019)*, the use of hydrogels may reduce and control irrigation in water conservation. While it is clear that even countries with large economies have been affected by food supply problems that have arisen due to the pandemic, there is even more uncertainty about how the food shortage will affect the whole world. There is an obligation to increase water optimization in agriculture and create digital systems with real-time monitoring capabilities and innovative resources. The use of hydrogel in agricultural activities is shown to be a solution to water shortages, especially in arid and semi-arid areas. The use of biodegradable hydrogel reduces the frequency of irrigation, increases the infiltration and water holding capacity of the soil, and eliminates the need for environmental protection. Therefore, ensuring the protection of water resources in agriculture (*Skrzypczak et al., 2020*).

Water stress, which is manifested by a lack of moisture in the plant root zone, causes early leaf senescence, low chlorophyll content, low seed yield, and lower fruit and flower formation. Hydrogels can have a positive effect on agriculture by reducing the effect of drought, which causes the formation of oxygen radicals. The addition of hydrogels to soil (in different soils and with different hydrogel amounts) increases the water holding capacity of the soil by 56% to 81% (*Saha, Gupt & Sekharan, 2021*). Water and land scarcity has increased significantly due to population growth, urbanization, pollution of resources, and the effects of climate change. Even though resources are scarce, the world needs to produce over 60% more food to feed nearly 10 billion people by 2050 (*Singh, 2022*). It is crucial to find a solution to the water shortage problem in agriculture. Unfortunately, agriculture is the area most affected by water scarcity. Drought and salt stress are some of the worst conditions for crop growth in arid and semi-arid regions due to lack of rainfall and fertilizer accumulation in the soil (*Esmaeili et al., 2021*).

The aim of the current research was to examine the effects of four levels of hydrogel (0%, 50%, 75%, and 100% (w/w)) on parsley (*P. crispum* (Mill.) Nyman) morphology, growth, and development under two levels of water deficit stress (50% and 100%). This is the first study that found that deficit irrigation combined with hydrogel application yielded better results than full irrigation.

## MATERIALS AND METHODS

### Experiment location
The study was conducted at the Siirt University Faculty of Agriculture, Department of Field Crops Tissue Culture Laboratory.

### Plant material
The plant material used in this experiment was the Aspuzu variety of parsley. This variety is small, slightly curved, oval shaped, striped, gray-green in color, strongly aromatic, and flourishes in regions with high humidity and temperate climate.

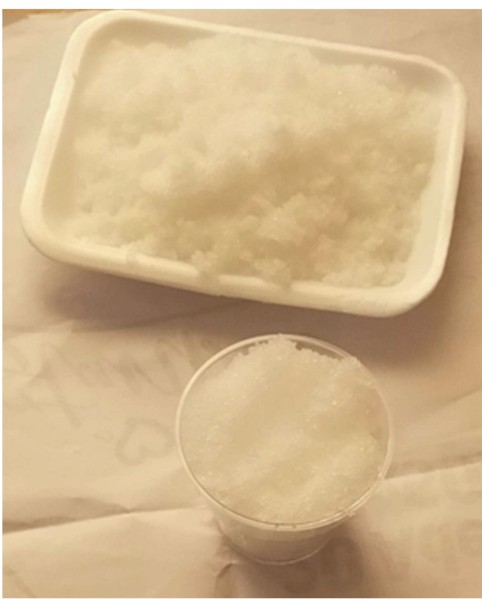

**Figure 1 Polyacrylate polymer used in the experiment.** The hydrogel (Polyacrylate polymer) is used at the experiment.

### Experimental treatments and design

The experiment, involving four different hydrogel doses and two different water levels were applied, was performed using a randomized block design with three replications in laboratory conditions. In total, three measurements were made in each block. The first measurement was made 15 days after planting, and the other two measurements were made 10 days apart. The following traits were measured: root width and length, leaf width and length, main stem length, and number of siblings.

## METHODS

The amount of water given to each block was determined according to the results of time-domain reflectometry (TDR) measurements. Distilled water was applied as irrigation water. Two water levels were applied as water treatments:

I1: 50%

I2: 100%.

The nutrients were dissolved in water and given irrigation. The plant growth medium was prepared by mixing the hydrogel at the determined rates in each pot. In the 0% treatment, only peat was included in the pots. In the 100%, the hydrogel was completely contained. Polyacrylate polymer (Fig. 1) was used as a hydrogel. Four hydrogel ratios were administered:

H1: 0%

H2: 50%

H3: 75%

H4: 100%

### Plant growth conditions

The plants were grown in 1-l pots filled with peat. The experiment was conducted as three replications with five pots in each replication and one plant per pot. All plant measurements were performed on three plants in each replicate.

## OBSERVATION

Plant growth measurements were first taken 15 days after planting and then at 10 day intervals. The length of the main root and overall root width were determined using a digital caliper. The leaf length was determined by separating the petiole from the main stem from three leaves of each plant using a caliper. The leaf width and length were also measured using a caliper in leaves. The main stem length was determined by measuring with a ruler from the root collar. Tillers that originated from the same root and were located around the main stem were counted.

### Statistical analysis

Data analysis was done using SAS 9.1 statistical software (SAS version 9.3; SAS Institute, Cary, NC, USA) according to the randomized complete blocks (RCBD) experimental design. Least significant difference (LSD) multiple comparison tests were used to compare the treatment means. The averages of the applications and the averages of each measurement in each application were calculated using Excel.

## RESULTS AND DISCUSSION

All evaluated traits were significantly affected by the different rates of hydrogel application and irrigation levels (Table 1). Plant development was completely stopped and all plants died 15 days after planting when the hydrogel level was at 100%. For this reason, no plants were measured in the H4 application, even in the first measurement. The H3 application had lower values compared to the H2 and H1 applications. The H2 application had the highest values in all measurements, while the H1 application shared the same statistical group with the H2 application except in leaf width and height and main stem length. It has been noted that hydrogel application enhances plant growth, but higher rates of hydrogel did not induce better results. We concluded that the addition of 50% hydrogel to the plant growth medium had a positive effect on the growth. It has also been found that the 50% water application (drought treatment) resulted in higher values than the full water application (Table 1).

Table 2 shows that the highest mean value (3.6) in all measurements was obtained from a 50% irrigation application (I1). According to the averages of different irrigation applications, full irrigation (I2) was in second with an overall mean value of 2.4. These results show a higher performance under 50% irrigation compared to 100%. Additionally, according to the hydrogel averages, the highest mean value (6.2) was obtained in all measurements of 50% hydrogel application (H1) and the lowest value (0.0) was obtained from plants with 100% hydrogel application (H3). The effects of hydrogel and water applications were also significant on root length. When the effects of irrigation and hydrogel applications on each measured trait were examined separately, it was found that

**Table 1 Effects of hydrogel levels and irrigation on parsley morphological parameters.**

| Applications | | Root length (cm) | Root width (cm) | Leaf length (cm) | Leaf width (cm) | Main stem length (cm) | Tillers (number) |
|---|---|---|---|---|---|---|---|
| Hydrogel | H1 (0%) | 6.5b | 0.3b | 2.8b | 2.2a* | 14.1a* | 2.7b |
| | H2 (50%) | 7.5a* | 2.2a* | 3.4a* | 2.5a* | 15.0a* | 3.3a* |
| | H3 (75%) | 2.6c | 0.2bc | 0.6c | 1.1b | 3.4b | 1.0c |
| | H4 (100%) | 0.0d | 0.0c | 0.0d | 0.0c | 0.0c | 0.0d |
| | Std. Error | 0.14976 | 0.05527 | 0.13228 | 0.08457 | 0.44386 | 0.16666 |
| *p* value | | *p* < 0.0001 | *p* < 0.0001 | *p* < 0.0001 | *p* < 0.0001 | *p* < 0.0001 | *p* < 0.0001 |
| Irrigation | $I_1$ (50%) | 4.9a* | 1.1a* | 2.2a* | 1.9a* | 9.4a* | 2.0a* |
| | $I_2$ (100%) | 3.4b | 0.3b | 1.2b | 1.0b | 6.9b | 1.5b |
| | Std. Error | 0.105 | 0.039 | 0.093 | 0.059 | 0.314 | 0.118 |
| *p* value | | *p* < 0.0001 | *p* < 0.0001 | *p* < 0.0001 | *p* < 0.0001 | *p* < 0.0001 | *p* < 0.0001 |

Notes:
* Significant at $p < 0.001$, all analyses were performed *via* three replicates.
It was concluded that the hydrogel addition of 50% to the plant growth medium had a positive effect on the growth of the plants. It has also been found that 50% water application (drought treatment) had higher values than full water application.

**Table 2 The average values of each application.**

| Applications | | Root length | Root width | Leaf length | Leaf width | Main stem length | Tillers | Mean | |
|---|---|---|---|---|---|---|---|---|---|
| Irrigation | 50% | 4.9 | 1.1 | 2.2 | 1.9 | 9.4 | 2.0 | 3.6 | $I_1$ |
| | 100% | 3.4 | 0.3 | 1.2 | 1.0 | 6.9 | 1.5 | 2.4 | $I_2$ |
| Hydrogel | 0% | 6.5 | 0.3 | 2.8 | 2.2 | 16.2 | 2.7 | 5.1 | $H_0$ |
| | 50% | 7.5 | 2.2 | 3.4 | 2.5 | 18.2 | 3.3 | 6.2 | $H_1$ |
| | 75% | 2.6 | 0.2 | 0.6 | 1.1 | 4.3 | 1.0 | 1.6 | $H_2$ |
| | 100% | 0.0 | 0.0 | 0.0 | 0.0 | 0.0 | 0.0 | 0.0 | $H_3$ |

Note:
Hydrogel and water application rates and averages were found to have effects on agronomic properties.

50% irrigation application (I1) resulted in the highest average root length (4.9 cm), while 100% irrigation (I2) resulted in the shortest root length (3.4 cm). When the effects of hydrogel applications were examined on root length, the highest value (7.5 cm) was found in the 50% hydrogel application (H1). Irrigation applications and hydrogel applications had different effects on root width. Of the different irrigation applications, the 50% application had the highest root width value (1.1 cm) while 100% irrigation application had the lowest average (0.3 cm). Of the different hydrogel applications, the highest root width value (2.2 cm) belonged to the H1 (50% hydrogel) application, while the lowest value recorded (0.0 cm) belonged to the full hydrogel application (100% hydrogel). With 50% irrigation, the highest leaf length and width values were 2.2 cm and 1.9 cm, respectively, while in the H1 application (50% hydrogel), the highest leaf length and width values were 3.4 cm and 2.5 cm, respectively. The H1 application resulted in higher values than the other hydrogel applications, with 18.2 cm main plant height and 3.3 tillers per plant. The 50% irrigation application had the highest values with 9.4 cm main plant height and 2.0 tillers per plant in comparison with the other irrigation applications.

**Table 3 Effects of hydrogel and irrigation interactions on parsley morphological parameters.**

| Applications | | Root length (cm) | Root width (cm) | Leaf length (cm) | Leaf width (cm) | Main stem length (cm) | Number of tillers (Number) |
|---|---|---|---|---|---|---|---|
| Hydrogel × Irrigation | H1 × I$_1$ | 6.5b | 0.3bc | 2.8b | 2.2b | 14.1b | 2.7b |
| | H2 × I$_1$ | 9.9a* | 3.8a* | 4.9a* | 3.6a* | 18.0a* | 4.3a* |
| | H3 × I$_1$ | 3.2d | 0.2cd | 0.9d | 1.6c | 5.5d | 1.0c |
| | H4 × I$_1$ | 0.0f | 0.0d | 0.0e | 0.0e | 0.0e | 0.0d |
| | H1 × I$_2$ | 6.5b | 0.3bc | 2.8b | 2.2b | 14.1b | 2.7b |
| | H2 × I$_2$ | 5.1c | 0.5b | 1.8c | 1.3c | 12.0c | 2.3b |
| | H3 × I$_2$ | 2.0e | 0.2cd | 0.9d | 0.5d | 1.3e | 1.0c |
| | H4 × I$_2$ | 0.0f | 0.0d | 0.0e | 0.0e | 0.0e | 0.0d |
| | Std. Error | 0.21180 | 0.07817 | 0.18708 | 0.11960 | 0.62771 | 0.23570 |
| p value | | p < 0.0001 | p < 0.0001 | p < 0.0001 | p < 0.0001 | p < 0.0001 | p < 0.0001 |

Notes:
* Significant at $p < 0.001$, all analyses were performed *via* three replicates.
When the interactions of hydrogel and irrigation regimes were examined, it was determined that the highest measurement values in all measurements were obtained from the 50% hydrogel and 50% water application.

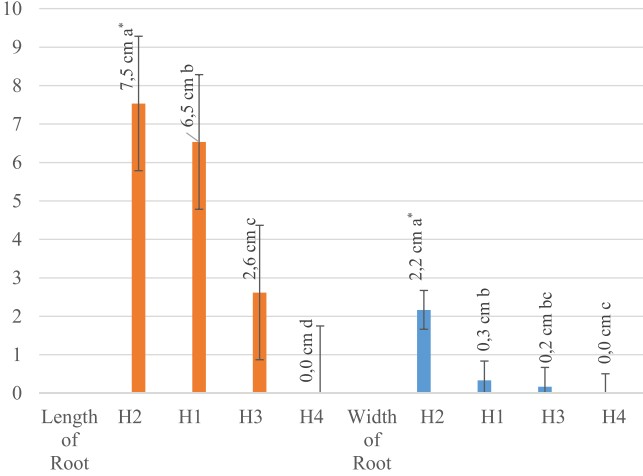

**Figure 2 Effect of hydrogel and irrigation interactions on root length and width (centimeter-cm, *significant at $p < 0.001$, all analyses were performed *via* three replicates).** It is seen that the highest measurement in both root length and root width values was obtained from 50% hydrogel application.

When the interactions of the hydrogel and irrigation regimes were examined, it was determined that the highest values across all measurements were obtained from the 50% hydrogel and 50% water application followed by the H1 × I1 and H1 × I2 groups (Table 3).

Since no measurements were recorded in the H4 × I1 and H4 × I2 interaction groups, it was noted that these interaction groups took last place (Table 3). Control (0% hydrogel) and 50% hydrogel applications differed significantly from other applications in terms of shape, and the highest value 50% hydrogel application ranked first in all applications (Figs. 2–4).

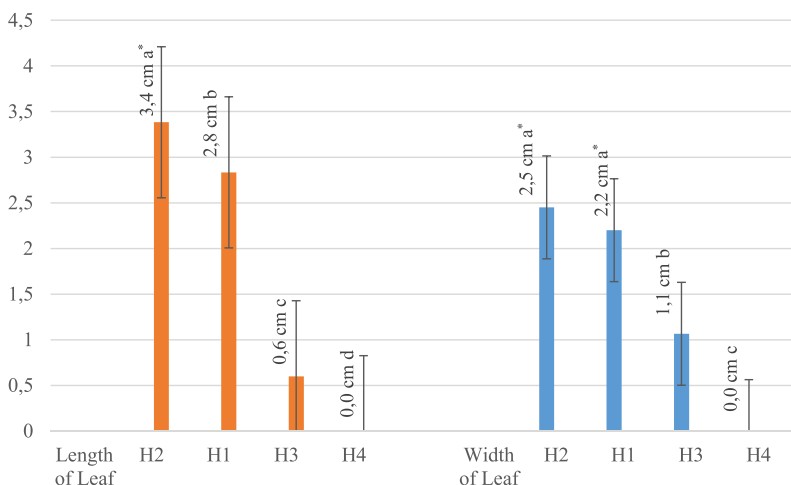

**Figure 3 Effect of hydrogel and irrigation interactions on leaf length and width (centimeter-cm, *significant at *p* < 0.001, all analyses were performed *via* three replicates).** In the leaf length and width measurements, as in the root length and root width measurements, the highest values were obtained from 50% hydrogel application.

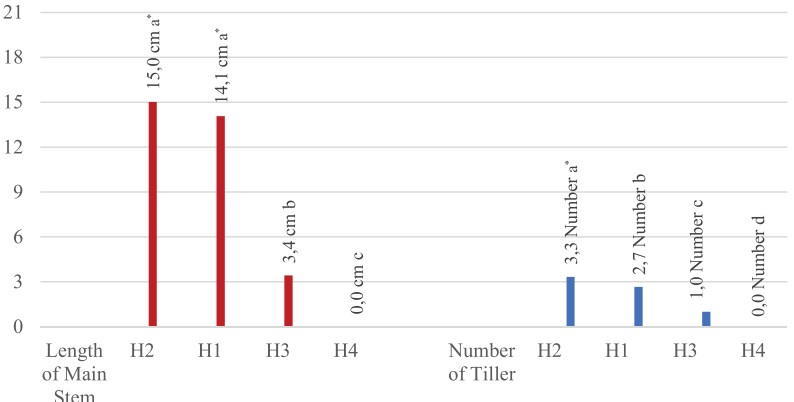

**Figure 4 Effect of hydrogel and irrigation interactions on main stem length (centimeter-cm) and number of tillers (Number) *significant at *p* < 0.001, all analyses were performed *via* three replicates.** It is understood that the main stem length values and the values of tillering numbers obtained in 50% hydrogel environment are better than other hydrogel applications.

Pictures of the measured parsley plants are presented in Fig. 5. The attachment of hydrogel to the roots of the plants can be clearly seen. It was determined that high doses of hydrogel applications stopped plant root development. At the same time, it was determined that high water application (H3 × I2) prevented the roots from fringing and developing in depth. Although vertical root development in all plants varied according to the applications, in Fig. 1 we show that the fringe rooting changed according to irrigation and hydrogel ratios. Adhesion of hydrogels to the fibrous roots of plants with 50% hydrogel was an indication that this plant group had grown higher than the other treatments (Fig. 5).

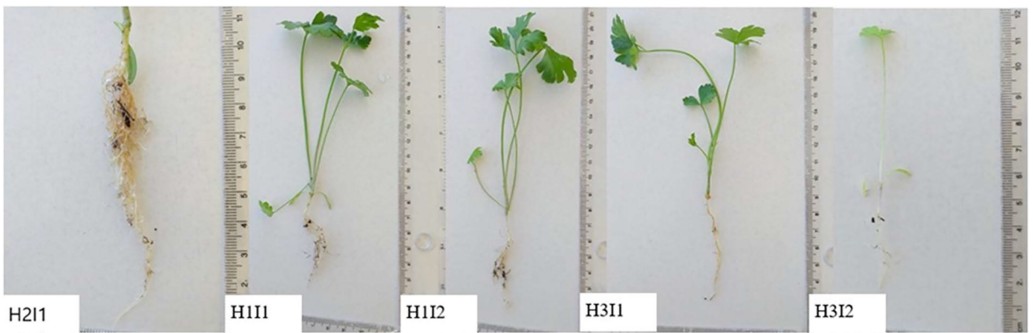

**Figure 5  Parsley seedlings under different hydrogel and irrigation treatments.** Adhesion of hydrogels in the fibrous roots of plants with 50% hydrogel is an indication that this plant group has grown higher than the other treatments.

According to *Liu, Wang & Kapteijn (2020)*, the use of hydrogel increased work efficiency, but most importantly, it had a positive effect on the growth of plants and provided net economic benefits. We determined that a 50% application provided water economy and sustainable water use, similar *Liu, Wang & Kapteijn's (2020)* findings. Similarly, *Liu, Wang & Kapteijn (2020)* obtained positive results in the water infiltration and capacity of the soil when hydrogel powders were mixed into sandy soils. Similar findings were found in studies conducted on different crops such as wheat (*Shooshtarian, Abedi-Kupai & TehraniFar, 2012*), rice (*Nazarli & Zardashti, 2010*), millet (*Singh, 2012*; *Keshavars, Farahbakhsh & Golkar, 2012*), and peanut (*Langaroodi et al., 2013*). According *Rajanna et al. (2022)*, hydrogels increased the yields of soybean and wheat plants by 20.6% and 52.7%, respectively, compared to the control.

The aqueous environment around the roots of the plant, reduces the effects of drought by preventing moisture loss in the soil under dry conditions and acts as a buffer, especially in the early stages of plant development (*Borivoj, Rak & Bubenikova, 2006*). Hydrogel application under water stress conditions increased plant height, stem diameter, vegetative period, and leaf length along with overall growth of maize plant (*Albalasmeh et al., 2022*; *Radian, Ichwan & Hayat, 2022*; *Watcharamul et al., 2022*; *Prisa & Guerrini, 2022*). The control group in our experiment had lower plant growth values than the plants with 50% hydrogel application, and our results were in a similar range with previous reports. *Shankarappa et al. (2020)* found that hydrogel application increased pod and seed yield in lentils. It has been determined that hydrogel applications have a positive effect on the growth of *Raphanus sativus* and *Phaseolus vulgaris* vegetables in arid conditions and enable these plants to have longer life spans (*Durpekova et al., 2022*). Hydrogel applications increased leaf area, as well as the yield of leafy vegetables by 35% to 60% and encouraged the development of *Brassica rapa* subsp *chinensis* var *parachinensis* (*Zhu et al., 2022a*, *2022b*). In our experiment, the increase in leaf and root size, main stem length, and number of tillers in parsley were found to be higher than in *Shankarappa et al. (2020)*, although the results were similar. The use of hydrogel also improved the quality and properties such as color, shape, and biomass (*Shubhadarshi & Kukreja, 2020*). According to *Elbarbary & Ghobashy (2017)* and *Ibrahim, Abd-Eladl & Abou-Baker (2015)*, hydrogel

used in corn production increased cob quality, as well as earlier and higher yield in the plants. *Ghasemi & Khushkhui (2008)* reported a similar increase in the number of flowers, root/shoot ratio, and plant habit of the chrysanthemum. The leaf area and calyx yield of *Hibiscus sabdariffa* L. (*Besharati et al., 2022*), fruit quality, and vegetative growth of mango (Shelly cv.) (*Alshallash et al., 2022*) increased even under water stress conditions due to hydrogel applications.

In our study, the use of hydrogel had a positive effect on all parsey measurements, which was in line with the findings of *Shubhadarshi & Kukreja (2020)*. *Song et al. (2020)* determined that hydrogels not only enhanced the soil water holding capacity but also increased the saturated hydraulic conductivity, water holding curve, and nutrient holding capacity. It has been determined that hydrogels increase soil moisture capacity and support the development of plants and increase yield (*Boatright et al., 1997*; *Davies et al., 2000*). In arid conditions, hydrogels also increased the life plants' span by approximately 9–14 days (*Song et al., 2020*; *Shankarappa et al., 2020*). The negative effect of full hydrogel application (H4) might be due to limited oxygen availability in the root zone. *Demitri et al. (2013)* achieved similar results to our findings using plants in a 100% hydrogel environment, and stated that powder or granular hydrogel mixed with soil swells when it finds water, creating air spaces in the soil at the same time. It was determined that hydrogel applications on tomato vegetables grown under two different water conditions in greenhouse conditions increased plant growth, plant height, stem diameter, and yield values (*Madramootoo et al., 2023*). According *Nassaj-Bokharaei et al. (2021)*, hydrogel application provided 22–45% more tomato plant growth under water stress. Another study carried out in tomato (*Demitri et al., 2013*) determined that hydrogel-applied plants did not need additional irrigation compared to the control group. It has been determined that the use of hydrogel is economical (*Chaudhary et al., 2020*). In these conditions, hydrogels can be used in agricultural production to reduce water scarcity (*Shubhadarshi & Kukreja, 2020*; *Das et al., 2021*; *Louf et al., 2021*). Hydrogels that create a water source in the root zone also increase the usable field capacity, increase plant growth and yield, and reduce the production cost. We can say that one of the reasons for the contribution of hydrogels to plant growth is that they prevent nutrients from being drained by irrigation water, as *Song et al. (2020)* stated. According to their findings, the leaching of N, total P, and available K in both nitrate and ammonium form with irrigation water is prevented by the use of hydrogels. According to *Shubhadarshi & Kukreja (2020)*, hydrogels are not affected much by salinity. One of the biggest advantages of hydrogels is that they provide water to the root zone of the plant in a controlled way. Agricultural production is severely, and at times fatally, affected by drought. Under dry conditions, the application of fertilizer to plants is also negatively affected, which significantly reduces plant yield (*Ashraf, Ragavan & Begam, 2021*).

If the main priorities are to keep enough water in the soil in the plant's root zone, ensure plant development without damage, and obtain high yields under various drought conditions, hydrogels may be a solution. Hydrogels ensure that there is constant moisture in the root regions of the plant. The water uptake of plants from hydrogels accelerates when moisture loss occurs in soil due to evapotranspiration. When re-watering, the

hydrogels swell again according to their capacity. If the development and yield deficiencies of plants due to insufficient soil moisture under completely arid or semi-arid regions are eliminated as a result of the use of hydrogels, hydrogels should be considered as one of the solutions for future drought conditions. As *Ashraf, Ragavan & Begam (2021)* indicated, the difficulties experienced worldwide during the pandemic has shown us that such promising solutions need to be multiplied and further developed.

When adequate irrigation is not available, the nutrients in the soil cannot be supplemented. However, hydrogels also contribute significantly to plant nutrition by containing the nutrient solution (*Das et al., 2021*). Hydrogels that hold the nutrient solution have a direct positive effect on ion uptake by preventing the ions from draining (nutrient leakage) from the soil (*Song et al., 2020*) and also by dissolving ions with water. Adventitious roots that provide nutrient uptake increase when there is sufficient moisture in the soil. The continuous presence of water in the root environment with hydrogels provides the formation of lateral roots rather than deep roots and thus increases the nutrient (ion) uptake. The lack of water in the rhizosphere causes an increase in salt concentration. In saline environments, the ion uptake in leaves decreases rapidly (*Suthar et al., 2019*). However, hydrogels constantly retain moisture in the rhizosphere and indirectly prevent the increase in salinity density.

At the beginning of the twentieth century, water management became one of the biggest challenges in the world, especially in arid and semi-arid regions. Unfortunately we will face a 50% increased water use as of 2030. Since more than 70% of the water used in agricultural irrigation is usable/clean water, water conservation has become an urgent issue. One of the possible solutions to this problem is the use of hydrogels in soil management practices. It has been reported that the hydrogel returns water to the dried soil after storing it in its body.

This, when combined with predicted drought and rapidly increasing world population, indicates that a grave and chaotic future is ahead. There is an urgent need to eliminate drought and the negative effects caused by drought.

## CONCLUSION

The recent epidemic and intensifying drought pressure brought on by climate change has demonstrated the significance of the world's food supply. The utilization of hydrogels, a chemical component that can be used in water shortage situations and to combat the anticipated water scarcity, was investigated in this study. We utilized parsley as our model organism. Four distinct hydrogel concentrations (0%, 50%, 75%, and 100%) were employed, along with two different water treatments (50% and 100%). We looked at root breadth and length, leaf width and length, main stem length, and tiller count. The best outcomes were obtained by applying 50% hydrogel to the plants, but no improvement was seen in those with 100% hydrogel concentration. Results from a 75% hydrogel application were shown to be better than those from a 100% hydrogel application, but worse than those from a 0% hydrogel application. All plant growth parameters were higher under 50% hydrogel circumstances (water restriction) compared to the plants receiving 100% (full irrigation) water application. In all measured parameters, the best results in the values of

irrigation applications were obtained from 50% irrigation. When hydrogel applications were examined, it was determined that 50% hydrogel application performed best in all measured traits, but shared the same statistical group with the 0% hydrogel application in terms of leaf width and main plant height. In the analysis of water and hydrogel interactions, the results became more pronounced. All of the values belonging to the best statistical group were obtained in the H2I1 plant (50% hydrogel × 50% irrigation) interaction group. The study's findings indicate that hydrogels have great potential for reducing the impacts of drought on vegetable crops.

### Funding
The authors received no funding for this work.

### Competing Interests
The authors declare that they have no competing interests.

### Author Contributions
- M. Zeki Karipçin conceived and designed the experiments, performed the experiments, analyzed the data, prepared figures and/or tables, authored or reviewed drafts of the article, and approved the final draft.

### Data Availability
The raw measurements are available in the Supplemental File.

### Supplemental Information
Supplemental information for this article can be found online at http://dx.doi.org/10.7717/peerj.15105#supplemental-information.

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
