# Peer review of "Hydrogels improved parsley (Petroselinium crispum(Mill.) Nyman) growth and development under water deficit stress"

_PeerJ, doi:10.7717/peerj.15105_

## Round 0.1 · original submission · Major Revisions

Dear Professor Karipçin:

Thank you for submitting your manuscript “Impact of hydrogel polymer on parsley (Petroselinium crispum (Mill.) Nyman) growth and development‶ for publication in PeerJ – Life and Environment.

Based on input of the reviewers, I have decided that your manuscript may become acceptable for publication. It is important that you carefully revise the manuscript in response to the comments below and provide a clear response regarding the revisions you have made. Please be sure to highlight the changes to your manuscript within the document by using the track changes mode in MS Word.

Based on these revisions, a final decision will be rendered regarding acceptance.

Sincerely,
Professor Magdi Abdelhamid
Editor, PeerJ – Life and Environment

Reviewer 1 ·

Basic reporting

The results are questionable. Mainly the higher values of different traits in hydrogel as compared to 100% irrigation.

Materials and methods are not sufficient and not well documented

Statistical analysis is not sufficient.

Experimental design

Sufficient

Validity of the findings

Not sufficient

Additional comments

The results are questionable. Mainly the higher values of different traits in hydrogel as compared to 100% irrigation.

·

Basic reporting

Language is clear and concise. Grammatically correct and flow of language is good. The structure of the manuscript meets the required standard. Literature has been cited extensively. Most of the figures and tables are relevant and of good resolution.

However, Figure no.1 does not have a contrasting background and too unicolor for perception. I suggest improving the quality of this picture. Resolution of Figure no.5 is compromised and the markings on the scale are not visible clearly. I recommend improving the resolution of this figure.
Introduction of the paper is off to a strong start and sets a good premise of the problem (drought related stress on plant) and solution to it (use of hydrogel).

The addition of hypothesis would strengthen the introduction of the paper, considering that the author is studying the effect of hydrogel in normal and drought irrigation conditions. The novelty of this article which is mentioned in the last line of the discussion “This study is the first study that deficit irrigation with hydrogel application yielded higher than full irrigation”., should ideally be in the last para of introduction section. I believe this would strengthen the article’s hypothesis.

Experimental design

The experimental design of this paper is very good. I commend the author for a detailed and well written explanation of experimental design. This is very much within the scope of the journal.

The study is focused on application of hydrogel and its impact on plant growth in full irrigation and drought irrigation condition. The research question is well posed, defined and meaningful.

The investigation is relevant and statistically sound and reproducible.

However, I think there needs to be more analysis on what happens to the hydrogel in the soil after the plant growth cycle, considering its synthetic in nature. The water retention capacity of this hydrogel in the soil needs to be discussed more explicit and in detail. The influence of salinity and other ions on the retention capacity of this hydrogel is not discussed and should be considered in the analysis and discussion. This would strengthen the discussion and conclusion session.

Validity of the findings

All underlying data have been provided; they are robust, statistically sound, & controlled.

Additional comments

Good article, statistically sound, addressing the comments on the properties of hydrogel would strengthen the discussion of the manuscript.

Reviewer 3 ·

Basic reporting

no comment

Experimental design

no comment

Validity of the findings

no comment

Additional comments

Comments to the Author

1. Please modify the keywords. Delete one of the keywords "Drought" or "Deficit irrigation".
2. In the abstract results in quantitative terms should be described.
3. Line 157 and 164; in tables 1 and 2 title’s "morphological properties" should be "morphological parameters".
4. Line 172-174; in figure legends 2, 3, and 4, SI units are required for each parameter.
5. Line 185; in figure 5 title's "applications" should be "treatments".
6. In the conclusion author did not highlight the interaction [Hydrogel] * plant growth parameters.

---

## Round 0.2 · Major Revisions

This manuscript is an interesting and potentially important topic, and the standard of presenting the research is good. However, there are many notes that should be considered before accepting this manuscript as follows:

- The title should represent the article's content and facilitate retrieval in indices developed by secondary literature services. A good title (i) briefly identifies the subject, (ii) indicates the purpose of the study, and (iii) gives important and high-impact words early.

- The abstract must be completely self-explanatory and intelligible in itself. It should include the following: 1. Reason for doing work, including rationale or justification for the research; 2. Objectives and topics covered; 3. Brief description of methods used. If the paper deals mainly with methods, give the basic principles, range and degree of accuracy for new methods; 4. Results; 5. Conclusions.

- Discussion: You need to rewrite the discussion and include recent references published in 2022 and 2021.

·

Basic reporting

No comments

Experimental design

No Comment

Validity of the findings

No comment

Additional comments

All scientific comments have been addressed.

Reviewer 3 ·

Basic reporting

no comment

Experimental design

no comment

Validity of the findings

no comment

Additional comments

no comment

Reviewer 4 ·

Basic reporting

Rewrite the aim of the paper
Add updated references in lines 95-108
Check the english grammar

Experimental design

Rewrite the aim of the paper
Rewrite the design of the experiment to be more clarify
Check the english grammar

Validity of the findings

Add legend at the end of figure to write the number of replications and statistical analysis
In Table 1 and 3 add legend about number of replications and statistical analysis of the letters writing in the table
Check the english grammar

---

## Round 0.3 · accepted · Accept

Thank you for addressing all of the reviewers' comments.

The manuscript is ready for publication.